# The Na⁺ leak channel NALCN controls spontaneous activity and mediates synaptic modulation by α2-adrenergic receptors in auditory neurons

**Tenzin Ngodup[1†], Tomohiko Irie[2†], Seán P Elkins[1], Laurence O Trussell[1]\***

[1]Oregon Hearing Research Center and Vollum Institute, Oregon Health & Science University, Portland, United States; [2]Department of Physiology, Kitasato University School of Medicine, Sagamihara, Japan

**\*For correspondence:**
trussell@ohsu.edu

[†]These authors contributed equally to this work

**Competing interest:** The authors declare that no competing interests exist.

**Abstract** Cartwheel interneurons of the dorsal cochlear nucleus (DCN) potently suppress multisensory signals that converge with primary auditory afferent input, and thus regulate auditory processing. Noradrenergic fibers from locus coeruleus project to the DCN, and α2-adrenergic receptors inhibit spontaneous spike activity but simultaneously enhance synaptic strength in cartwheel cells, a dual effect leading to enhanced signal-to-noise for inhibition. However, the ionic mechanism of this striking modulation is unknown. We generated a glycinergic neuron-specific knockout of the Na⁺ leak channel NALCN in mice and found that its presence was required for spontaneous firing in cartwheel cells. Activation of α2-adrenergic receptors inhibited both NALCN and spike generation, and this modulation was absent in the NALCN knockout. Moreover, α2-dependent enhancement of synaptic strength was also absent in the knockout. GABA_B receptors mediated inhibition through NALCN as well, acting on the same population of channels as α2 receptors, suggesting close apposition of both receptor subtypes with NALCN. Thus, multiple neuromodulatory systems determine the impact of synaptic inhibition by suppressing the excitatory leak channel, NALCN.

## eLife assessment

This paper reports the **fundamental** discovery of adrenergic modulation of spontaneous firing through the inhibition of the Na⁺ leak channel NALCN in cartwheel cells in the dorsal cochlear nucleus. This study provides unequivocal evidence that the activation of α-2 adrenergic or GABA-B receptors inhibit NALCN currents to reduce neuronal excitability. The evidence supporting the conclusions is **exceptional**, the electrophysiological data is high quality, and the experimental design is rigorous.

## Introduction

The dorsal division of the mammalian cochlear nucleus (DCN) is a cerebellum-like structure in which principal cells integrate multimodal activity with primary auditory afferents, and these converging signals are thought to contribute to sound localization and sensitivity to sounds of interest (*May, 2000*; *Oertel and Young, 2004*). Cartwheel cells (CWCs) are cerebellar Purkinje cell homologs that potently control this convergence by gating the multimodal signals to postsynaptic fusiform principal cells. CWCs also receive inputs from various neuromodulatory systems, including noradrenaline (NA) (*Trussell, 2019*). We showed previously that agonists of α2-adrenergic receptors simultaneously halt spontaneous spike activity and enhance the strength of inhibitory signals to fusiform cells (*Kuo and*

*Trussell, 2011*). The mechanism for this paradoxical action depends on the effect of spontaneous activity on synaptic strength: by reducing ongoing presynaptic firing, recovery from synaptic depression ensues and allows CWCs to mediate stronger postsynaptic signals in the fusiform cells. A related mechanism was later identified in the action of oxytocin in hippocampal interneurons (*Owen et al., 2013*; *Tirko et al., 2018*). However, the mechanism by which α2-adrenergic receptors controlled such activity remained obscure. The most obvious candidate ion channel that could mediate such inhibition is the GIRK channel (G-protein-gated inwardly rectifying K⁺ channel) (*Lüscher et al., 1997*; *Arima et al., 1998*; *Li and van den Pol, 2005*; *Philippart and Khaliq, 2018*); however, attempts in our laboratory to implicate this channel failed.

The Na⁺ leak channel NALCN functions as a complex of four proteins: NALCN, FAM151a, UNC79, and UNC80 (*Lu et al., 2010*; *Ren, 2011*; *Cochet-Bissuel et al., 2014*; *Kschonsak et al., 2020*; *Xie et al., 2020*). This channel complex (here termed simply NALCN) contributes to the depolarizing drive for spontaneous firing in a wide variety of neurons, and functions in sensory, motor, and circadian pathways (*Nash et al., 2002*; *Lu et al., 2007*; *Lu and Feng, 2011*; *Xie et al., 2013*; *Lutas et al., 2016*; *Shi et al., 2016*). Accordingly, mutations in the NALCN subunit or its associated proteins are linked to a variety of human diseases (*Al-Sayed et al., 2013*; *Cochet-Bissuel et al., 2014*; *Chong et al., 2015*; *Bramswig et al., 2018*). A hallmark of this channel is its inhibition by extracellular $Ca^{2+}$; reduction of extracellular $Ca^{2+}$ enhances the current, permitting a ready assessment of NALCN's presence in neurons (*Lu et al., 2010*; *Ren, 2011*; *Philippart and Khaliq, 2018*; *Chua et al., 2020*). Recently, it was shown that neurons of the substantia nigra express NALCN, and that both dopamine and $GABA_B$ receptors strongly downregulated NALCN activity in this region (*Lutas et al., 2016*; *Philippart and Khaliq, 2018*). As global knockouts of NALCN die at birth (*Lu et al., 2007*), we generated a knockout mouse line specific to glycinergic neurons in order to test the role of this channel in auditory interneurons. We found that CWCs expressed functional NALCN, and its loss in NALCN knockouts led to cessation of spontaneous firing. Outward currents and inhibition of firing mediated by NA were eliminated in the knockouts; since NALCN generates inward excitatory currents, NA must act by inhibition of NALCN. Similar results were obtained for the $GABA_B$ agonist baclofen. Thus, NALCN serves multiple roles in auditory function, setting the pace of neuronal firing while also mediating electrical inhibition by multiple neuromodulators and enhancement of synaptic strength.

## Results
### NALCN and modulation of spike generation

In order to assess the role of NALCN in modulation of spontaneous firing of CWCs, we generated an NALCN conditional knockout (cKO) mouse by crossing a floxed *Nalcn* mouse line B6(Cg)-*Nalcn$^{tm1c(KOMP)Wtsi}$*/DrenJ with a GlyT2-cre line (the latter targeting Cre-recombinase to the *Slc6a5* locus), thus deleting NALCN from glycinergic neurons ('Materials and methods'). *Nalcn$^{-/-}$* mice were slightly smaller than wildtype mice (weight ratio, 4:3) but had normal hearing, as assessed by auditory brainstem responses (ABRs; *Figure 1—figure supplement 1*), and the morphology of their CWCs was characteristic of those of wildtype mice (*Figure 1—figure supplement 2*; *Bender and Trussell, 2009*). 'Wildtype' mice used here were either C57BL/6 or GlyT2-EGFP (*Zeilhofer et al., 2005*; *Kuo et al., 2012*; *Ngodup et al., 2020*) in which GFP is expressed in glycinergic neurons. CWCs were recorded using cell-attached mode to monitor baseline spontaneous firing and the inhibition of firing by NA. Except as noted below, all recordings were made in physiological $Ca^{2+}$ concentration (1.2 mM). In confirmation of previous studies in wildtype mice (*Kim and Trussell, 2007*; *Kuo and Trussell, 2011*), 71% of CWC exhibited spontaneous firing (mean firing rate, 17.74 ± 2.31 Hz) and bath application of 10 µM NA completely and reversibly eliminated such firing in every case (N = 8, *Figure 1A and B*). However, in experiments on NALCN cKO mice, CWC showed no spontaneous firing (*Figure 1C and D*; N = 6; difference from wildtype type, p<0.0005), and thus NA had no additional effects on spontaneous firing (mean firing rate, KO = 0 ± 0 Hz, NA = 0; N = 6). Thus, NALCN likely provides a tonic inward current necessary to drive spontaneous firing in CWC. These effects were not accompanied by changes in membrane input resistance (WT: 135.34 ± 21.53 MΩ [n = 18] vs KO: 132.19 ± 16.11 MΩ [n = 15], p=0.46, *t*-test).

Although NALCN supports spontaneous firing, it was not clear whether the effect of NA is mediated through NALCN or some other channel. To explore this problem, we first examined evoked

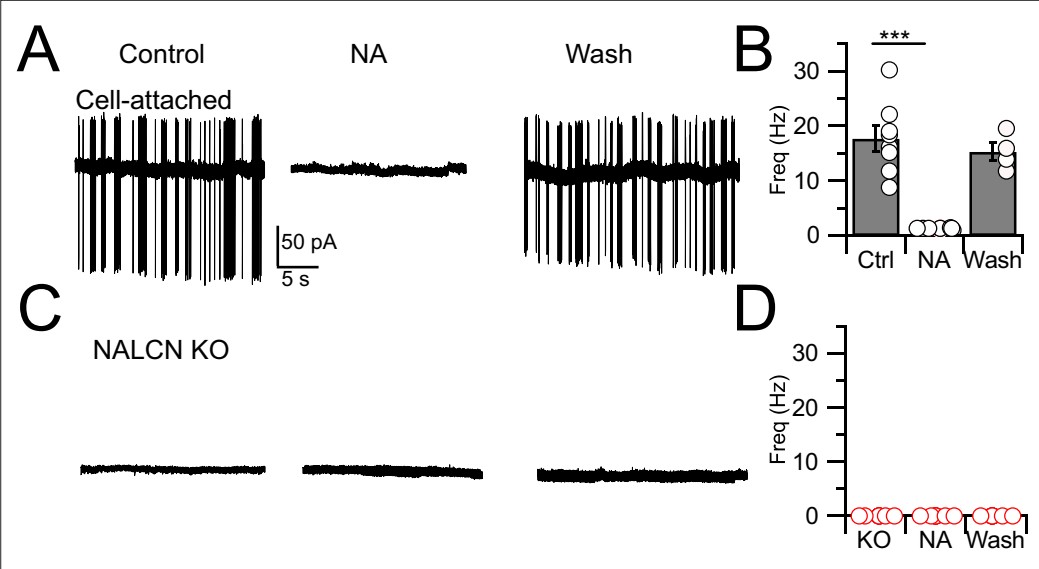

**Figure 1.** Spontaneous firing is absent in NALCN conditional knockout (cKO) mice. (**A**) Representative cell-attached recording from a spontaneously spiking cartwheel cell (CWC) in control. Noradrenaline (NA; 10 µM) applied to the bath in the middle trace, and washout on the right-most trace. (**B**) Spontaneous spike rate in control, NA, and washout, showing that NA eliminated spontaneous spiking in CWCs (n = 6 cells). (**C**) Representative trace showing absence of spontaneous firing in a CWC from a NALCN cKO mouse. NA had no effect on firing. (**D**) Summary of spontaneous firing and lack of NA effect in CWC from NALCN cKO mice (n = 5 cells). See *Figure 1—source data 1*.

The online version of this article includes the following source data and figure supplement(s) for figure 1:

**Source data 1.** Source data for *Figure 1B, D*.

**Figure supplement 1.** NALCN conditional knockout (cKO) mice show no difference in auditory brainstem response (ABR) from wildtype (WT).

**Figure supplement 1—source data 1.** Source data for *Figure 1—figure supplement 1B*.

**Figure supplement 2.** Biocytin-filled cartwheel cells (CWCs) from NALCN conditional knockout (cKO) mice.

firing in CWC and the effects of NALCN and NA. CWCs were recorded in current-clamp mode, and a family of negative and positive, 600-ms current pulses were delivered in 25-pA steps (*Figure 2A1*). In response to positive pulses, CWCs fire mixtures of simple (single) spikes and complex spikes, the latter containing $Na^+$ spike clusters driven by a $Ca^{2+}$-dependent current (*Kim and Trussell, 2007*). The timing of spikes during each current pulse was used to generate raster plots of firing (*Figure 2A2*), and the number of spikes at each current step was used to plot a frequency vs current intensity curve (*Figure 2E*). In the same set of cells, we then applied NA and repeated the current steps (*Figure 2B and E*). NA hyperpolarized the resting potential by –1.90 ± 0.24 mV and increased the current level required to initiate firing (rheobase, *Figure 2E*), indicating that NA reduced excitability. However, we also observed a suppression of the peak firing rate in NA; similar rundown of peak firing rate could be observed without change in rheobase as a result of whole-cell recording over this time period (see also *Kim and Trussell, 2007*), and thus frequency–intensity plots were normalized to peak intensity (*Figure 2F*). This manipulation revealed a clear shift to the right in the onset of firing (i.e., toward higher intensity) as a result of NA. Rheobase was defined as the current intensity at which spiking was 20% of maximum. Using this criterion, NA generated a significant increase in rheobase in response to current pulses (Ctrl, rheobase = 91.40 ± 11.35 pA, NA, 168.59 ± 14.89 pA, p=0.0011 [*t*-test]). Control experiments established that wash-in of NA with the α2 blocker idazoxan did not lead to a change in rheobase (Ctrl, rheobase = 95.11 ± 16.86 pA, NA + idazoxan, 105.88 ± 21.89 pA, N = 9, p=0.38 [*t*-test]), showing that the shift in rheobase was a genuine effect of α2 receptors (*Figure 2I and J*). This protocol was then applied to NALCN cKO mice (*Figure 2C, D, G, and H*). Although CWCs showed no spontaneous firing, they were able to respond to current steps with mixtures of simple and complex spikes, and peak firing rates did not differ from wildtypes (rheobase, Ctrl, 91.40 ± 11.35 pA; KO,

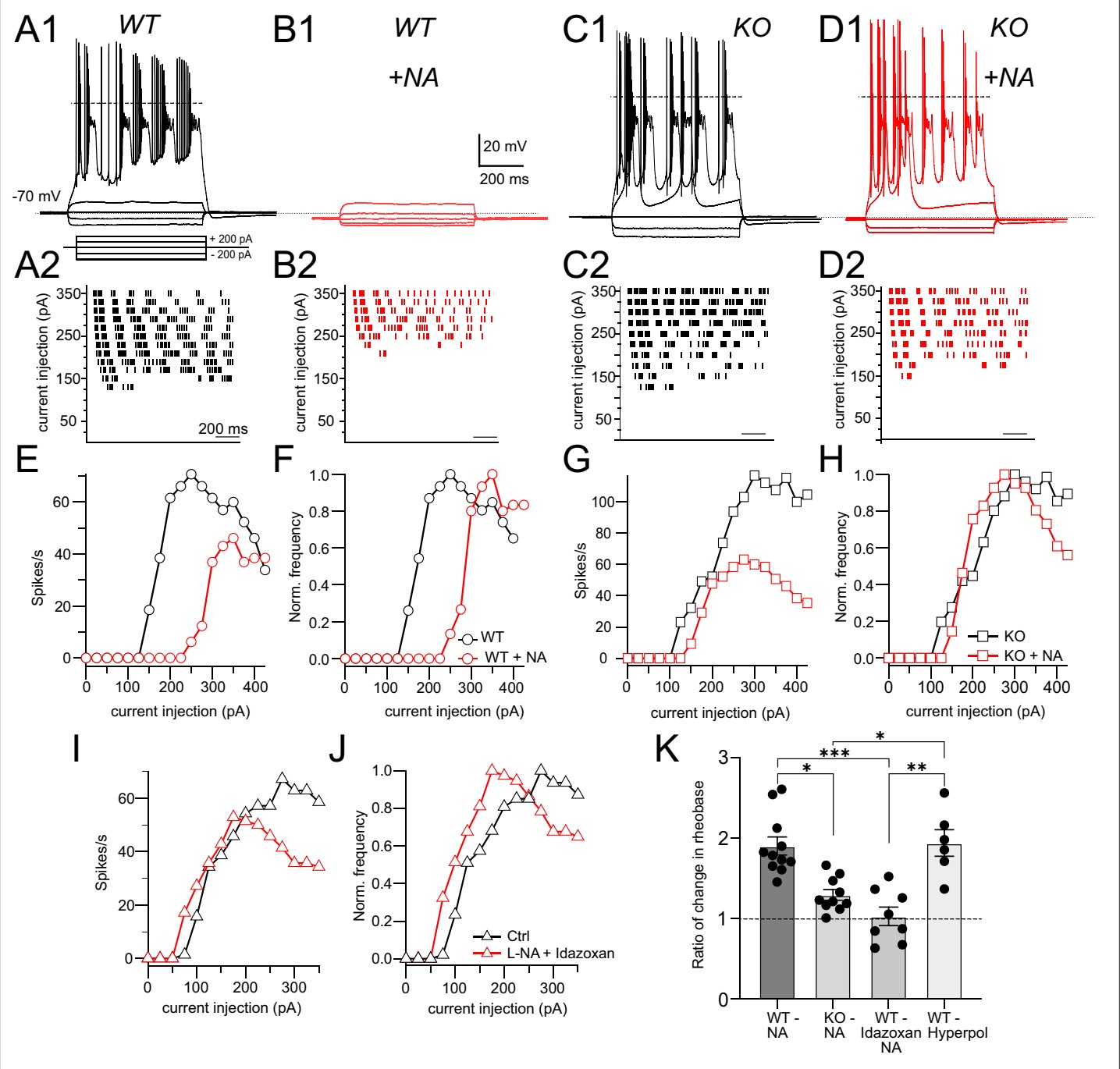

**Figure 2.** NALCN is required for noradrenaline (NA)-mediated shift in rheobase. (**A1**) Profile of voltage responses to current steps. Simple and complex spikes evoked by +200 pA. (**A2**) Raster plot of spike timings during different level of current injection. (**B1, B2**) Same cell as in (**A**) but in the presence of 10 μM NA (indicated by red traces and markers). (**C, D**) As in (**A, B**) but for recordings from a cartwheel cell (CWC) from a NALCN conditional knockout (cKO) mouse. (**E**) Spike rate calculated from data in (**A2**) and (**B2**) plotted as a function of current level. (**F**) Data from (**E**) normalized to peak firing rate. (**G**) Spike rate calculated from data in (**C2**) and (**D2**) plotted as a function of current level. (**H**) Data from (**G**) normalized to peak firing rate. (**I**), (**J**), raw and normalized frequency-intensity plots (respectively) for current responses from one WT cell in control solutions and in presence of 10 mM NA + 1 mM idazoxan. (**K**) Summary data (mean ± SEM) for ratio of change in rheobase of CWC from WT and KO mice with 10 μM NA, NA + idazoxan, and hyperpolarization conditions. Significance: *<0.05; **<0.01; ***<0.001. Dashed line in (**A1, C1, D1**) indicates 0 mV. See ***Figure 2—source data 1***.

The online version of this article includes the following source data and figure supplement(s) for figure 2:

**Source data 1.** Source data for ***Figure 2K***.

**Figure supplement 1.** Noradrenaline (NA) effect on rheobase and spiking is voltage dependent.

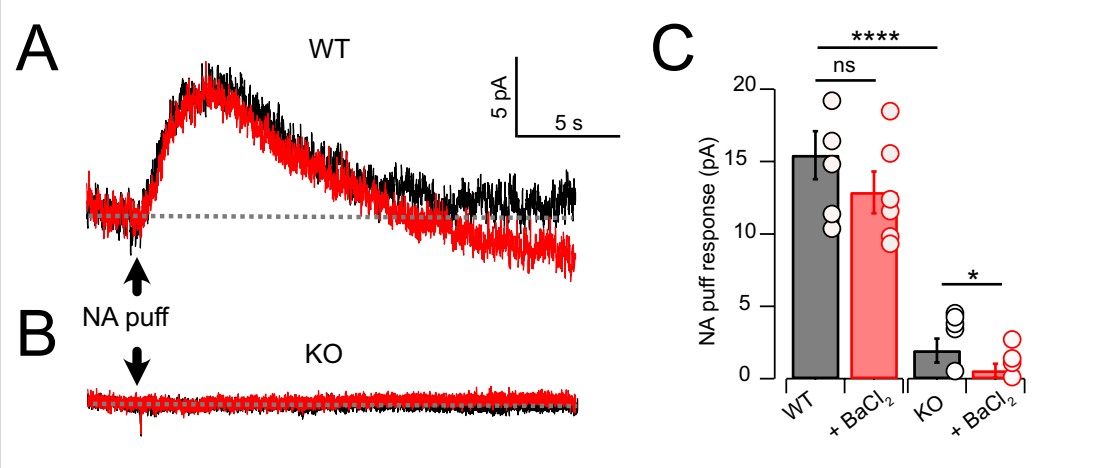

**Figure 3.** Noradrenaline (NA) evoked a $Ba^{2+}$-resistant outward current that required NALCN. (**A**) Outward current in response to NA puff (100 µM, 50 ms) in whole-cell voltage-clamp mode (– 65 mV) before (black) and after (red) block of GIRK channels by bath application of 100 µM $Ba^{2+}$. (**B**) As in (**A**), but from a cartwheel cell (CWC) from a NALCN conditional knockout (cKO) mouse. (**C**) Average data showing responses to NA puff in control conditions and after the block of GIRK channels from control (n = 5) and NALCN cKO mice (n = 6). In the KO, the NA response was markedly reduced and was $Ba^{2+}$ sensitive. Dashed line indicates initial current level. See **Figure 3—source data 1**.

The online version of this article includes the following source data for figure 3:

**Source data 1.** Source data for **Figure 3C**.

120.81 ± 17.53 mV, N = 11, p=0.08 [t-test]). However, unlike wildtype mice, the NALCN cKO mice showed no significant shift in rheobase with NA (KO, rheobase = 120.81 ± 17.53 pA, NA, 158.0 ± 12.16 pA, n = 11, p=0.064 [t-test]) (**Figure 2H**).

In order to test whether the effects of NA on rheobase and evoked spiking are primarily due to the small hyperpolarization induced by NA, we mimicked this effect of NA by injecting current to CWC to hyperpolarize the membrane by the same amount as induced by NA (approximately 2 mV), and then repeated the current injection protocol (**Figure 2—figure supplement 1**). Under these conditions, we were able to replicate the effects of NA on CWC rheobase and evoked firing (rheobase Ctrl = 70.88 ± 7.68 pA, hyperpolarized = 125 ± 12.90 pA, N = 6, p=0.0009). We then compared the ratio of the changes in rheobase in NA with respect to control under different conditions, also comparing to the relative effects of current injection. These comparisons confirmed that NA shifts rheobase dependent upon NALCN, and that this effect is likely caused by membrane potential hyperpolarization (**Figure 2K**: Ctrl = 1.90 ± 0.11, KO = 1.25 ± 0.05, idazoxan = 1.03 ± 0.11, hyperpolarized = 1.94 ± 0.16, p<0.0001, non-parametric Kruskal–Wallis; post hoc Dunn's test, Ctrl vs KO p=0.0086, Ctrl vs idazoxan p=0.0007, KO vs hyperpolarized p=0.02, idazoxan vs hyperpolarized p=0.0028).

### Outward response generated by suppression of NALCN

As cessation of spontaneous activity and elevation of rheobase by NA is likely associated with an outward, inhibitory current, we performed voltage-clamp experiments to determine the magnitude of that current in wildtype and KO mice, and its pharmacological sensitivity. Using a patch-pipette fill containing high $K^+$ ('Materials and methods'), cells were held at –65 mV and a 50-ms puff of NA was applied near the soma using a pressure-ejection pipette. Under these conditions, a slowly rising outward current was observed that decayed over a time course of 10–15 s (**Figure 3A**). As G-protein-coupled inward rectifier channels (GIRK) are known to mediate α2 receptor effects in some brain regions (**Williams et al., 1985**; **Aghajanian and Wang, 1986**; **Arima et al., 1998**; Li and **Li and van den Pol, 2005**; **Nimitvilai et al., 2017**), we tested the role of GIRK channels using the GIRK blocker $Ba^{2+}$. However, $BaCl_2$ (100 µM) had no significant effect on the amplitude of the NA-evoked current (**Figure 3A and C**) (Ctrl = 15.44 ± 1.66 pA; $BaCl_2$ = 12.86 ± 1.44 pA, N = 6, p=0.133). When this experiment was repeated in NALCN cKO mice, the NA-evoked current was only 12.5% of that observed in wildtype mice (**Figure 3B and C**). Although the amplitude of the NA-evoked current in the knockout was only a few pA, bath application of $BaCl_2$ nevertheless produced a statistically significant block

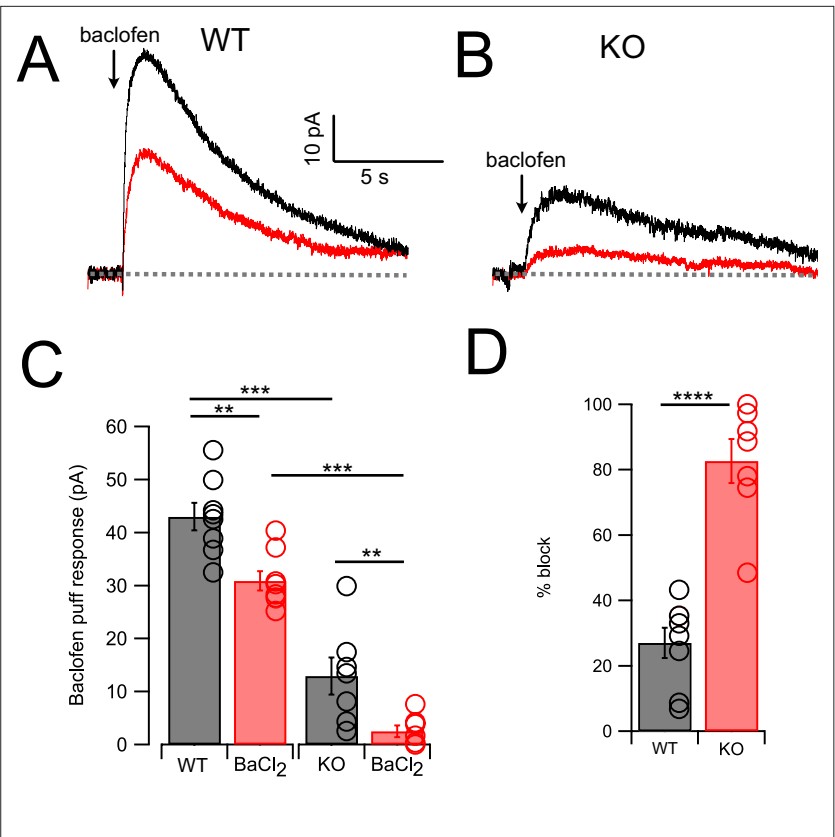

**Figure 4.** GABA_B receptors activate outward current mediated by NALCN and GIRK channels. (**A**) Traces from one neuron showing baclofen puff (100 µM, 50 ms) evoked outward current in control solution (black) and with 100 µM Ba²⁺ in bath (red). (**B**) As in (**A**) but for a cartwheel cell (CWC) from a NALCN conditional knockout (cKO) mouse. (**C**) Averaged data showing Ba²⁺ sensitivity of the noradrenaline (NA) response in wildtype (WT) and knockout (KO) tissue. Ba²⁺ produced a significant block in all cases, while KO CWC showed significantly smaller baclofen responses. (**D**) The degree of block by Ba²⁺ was significantly greater in KO mice, indicating that more of the baclofen response is mediated by GIRK channels after KO of NALCN. Dashed line indicates initial current level. See *Figure 4—source data 1*.

The online version of this article includes the following source data for figure 4:

**Source data 1.** Source data for *Figure 4C, D*.

---

of this small residual current, essentially eliminating all response to NA (KO = 1.93 ± 0.83 pA, BaCl₂ = 0.56 ± 0.4 pA, n = 8, p=0.05) (*Figure 3C*). Thus, the outward response to NA was only minimally due to activation of K⁺ current but rather was mediated largely by inhibition of a tonic inward current generated by NALCN. *Philippart and Khaliq, 2018* observed that GABA_B receptors suppressed NALCN current in substantia nigra, and so we asked whether these receptors might have a similar action in CWCs. In K⁺ filled cells, puffs of the GABA_B agonist baclofen generated outward currents similar to those observed with NA (*Figure 4A and C*). However, unlike NA, subsequent wash-in of BaCl₂ blocked 26.99 ± 4.60% of the baclofen response (Ctrl = 42.95 ± 2.58 pA, BaCl₂ = 30.90 ± 1.84 pA, N = 8, p=0.0014) (*Figure 4A, C, and D*), suggesting the partial involvement of GIRK channels. In NALCN cKO mice, the baclofen currents were significantly smaller than in wildtype (*Figure 4B and C*). Moreover, BaCl₂ in the KO had a much greater effect on the baclofen current compared to wildtype, blocking by 82.68 ± 6.71% (*Figure 4B–D*) (KO = 12.91 ± 3.51 pA, BaCl₂ = 2.57 ± 1.06 pA, N = 7, p=0.0089), indicating that after loss of NALCN, the remaining baclofen response was largely dependent on GIRK channels.

## Low Ca²⁺-induced inward current mediated by NALCN

NALCN current is enhanced by reduction of extracellular Ca²⁺ (*Lu et al., 2007*; *Lu et al., 2010*; *Ren, 2011*; *Philippart and Khaliq, 2018*; *Chua et al., 2020*), and in the presence of blockers of

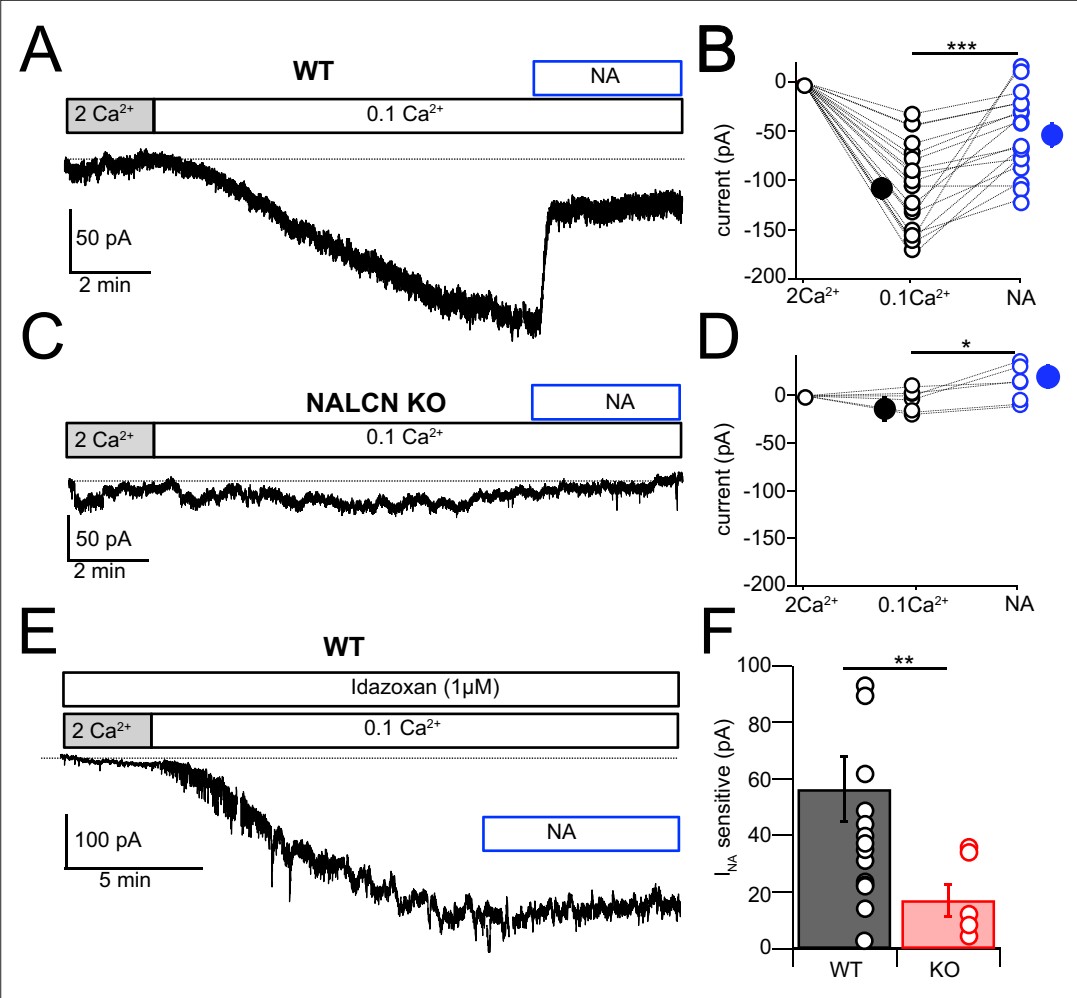

**Figure 5.** NALCN current evoked by Ca²⁺ reduction is inhibited by α2 receptors. (**A**) Shifting bath Ca²⁺ from 2 mM to 0.1 mM evokes a slow inward current that is then rapidly reduced by subsequent wash-in of 10 μM noradrenaline (NA). (**B**) Group data showing the magnitude of inward current shift in 0.1 mM Ca²⁺ (black) and the significantly smaller shift in 0.1 mM Ca²⁺ plus NA (blue). N = 18 cells. (**C**) Experiment as in (**A**) but for a cartwheel cell (CWC) from a NALCN conditional knockout (cKO) mouse. (**D**) As in (**B**), but for CWC from knockout (KO) tissue. N = 6 cells. (**E**) Experiment as in (**A**) but in continuous presence of 1 μM idazoxan. NA failed to block the low-Ca²⁺-evoked current. (**F**) The magnitude of inward current blocked by NA was significantly greater in wildtype (WT) compared to KO cells. All neurons voltage-clamped to –70 mV. Statistical significance: *p<0.05; **p<0.01; ***p<0.001. Extracellular solution contained TTX, NBQX, MK-801, strychnine, SR95331, and apamin. Dashed line indicates initial current level. See *Figure 5—source data 1*.

The online version of this article includes the following source data for figure 5:

**Source data 1.** Source data for *Figure 5B, D, F*.

voltage-dependent Na⁺, Ca²⁺, and K⁺ channels this effect is considered diagnostic for the presence of NALCN (*Philippart and Khaliq, 2018*). Recording in voltage clamp with the cocktail of intracellular and extracellular channel and receptor blockers described by *Philippart and Khaliq, 2018* (see 'Materials and methods,' and note that GIRK channels are also blocked here), an inward current of –107.61 ± 10.80 pA (N = 18) developed over several minutes upon shifting bath Ca²⁺ from 2 mM to 0.1 mM (*Figure 5A and B*). When NA was subsequently washed in, still in the presence of 0.1 mM Ca²⁺, the inward current was immediately reduced, falling by over half (–51.14 ± 9.44 pA, N = 18) (*Figure 5A, B, and F*). When similar experiments were performed in NALCN cKO mice, there was no change in holding current upon reduction in bath Ca²⁺, and only a minimal current response to NA (–4.78 ± 4.66 pA, N = 6) (*Figure 5C , D, and F*). In order to confirm that the reduction in inward

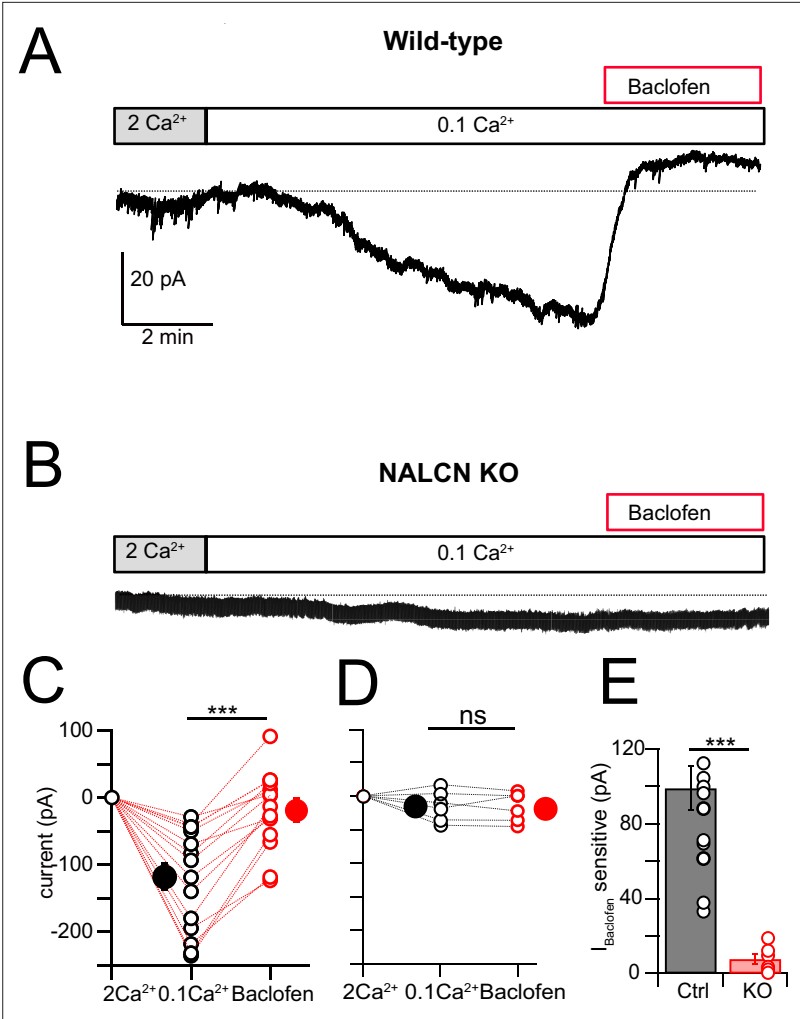

**Figure 6.** NALCN current evoked by $Ca^{2+}$ reduction is inhibited by $GABA_B$ receptors. (**A**) Shifting bath $Ca^{2+}$ from 2 mM to 0.1 mM evokes a slow inward current that is then rapidly reduced by subsequent wash-in of 10 μM baclofen. (**B**) Experiment as in (**A**) but for a cartwheel cell (CWC) from a NALCN conditional knockout (cKO) mouse. (**C**) Group data showing the magnitude of inward current shift in 0.1 mM $Ca^{2+}$ (black) and the significantly smaller shift in 0.1 mM $Ca^{2+}$ plus baclofen (red). N = 18 cells. (**D**) As in (**C**), but for CWC from knockout (KO) tissue. N = 6 cells. (**E**) The magnitude of inward current blocked by baclofen was significantly greater in wildtype (WT) compared to KO cells. All neurons voltage-clamped to –70 mV. Statistical significance: *p<0.05; **p<0.01; ***p<0.001. Extracellular solution contains TTX, NBQX, MK-801, strychnine, SR95331, and apamin. Dashed line indicates initial current level. See *Figure 6—source data 1*.

The online version of this article includes the following source data for figure 6:

**Source data 1.** Source data for *Figure 6C, D, E*.

current by NA was mediated by α2 receptors, we repeated the experiments in wildtype mice but in the presence of 1 μM idazoxan, and observed no effect on the inward current (Ctrl, 0.1 mM $Ca^{2+}$ = –112.52 ± 12.80 pA, NA = –108.35 ± 5.80 pA, N = 5, *Figure 5E*). Additional experiments were made using baclofen to activate $GABA_B$ receptors. Here, application of 10 μM baclofen almost completely eliminated the 0.1 mM $Ca^{2+}$-induced NALCN current (0.1 mM $Ca^{2+}$ = –117.75 ± 19.6 pA, baclofen = –18.80 ± 14.5 pA, N = 15, p=0.00018, *Figure 6A, C, and E*), while in knockout mice, neither 0.1 mM $Ca^{2+}$ nor baclofen altered the holding current (0.1 mM $Ca^{2+}$ = –14.48 ± 9.39 pA, baclofen = –15.90 ± 8.94 pA, N = 6, p=0.45, *Figure 6B, D, and E*). Altogether, these experiments support that NALCN is expressed in CWC and generates a $Ca^{2+}$-sensitive inward ionic current, which is suppressed by α2 and $GABA_B$ receptors.

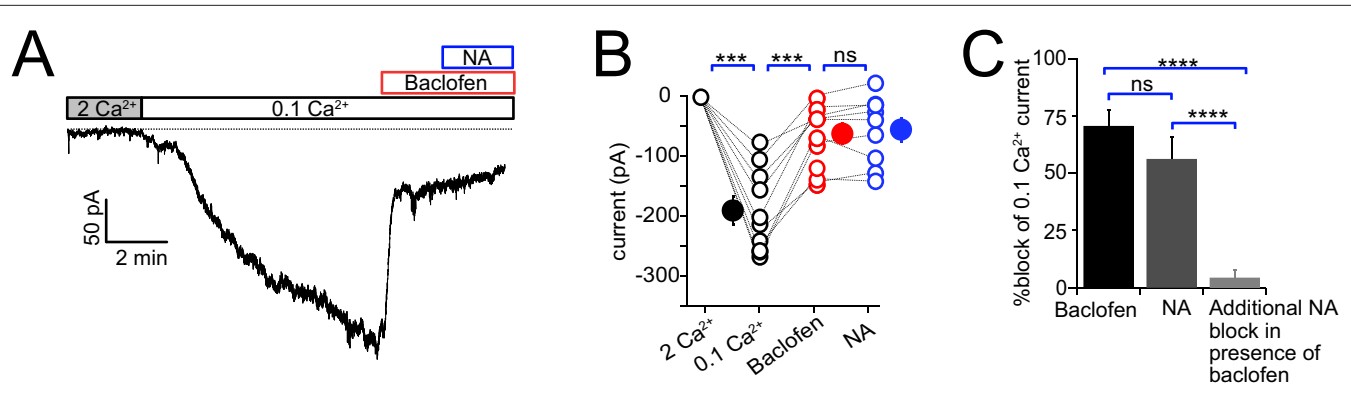

**Figure 7.** Baclofen and noradrenaline (NA) act on the same population of NALCN channels. (**A**) Representative example of NALCN current evoked by shift from 2 mM $Ca^{2+}$ to 0.1 mM $Ca^{2+}$, followed by bath application of baclofen (20 µM) and subsequent application of NA (10 µM) with baclofen still present. In the presence of baclofen, the rapid decline in inward current normally evoked by NA was absent. (**B**) Summary plot of NALCN current amplitude evoked by 0.1 mM $Ca^{2+}$ and after baclofen and subsequent NA application (wildtype [WT], N = 9). NA failed to produce a significant change after baclofen application. (**C**) Percentage block of 0.1 $Ca^{2+}$ current in baclofen, or NA alone, compared to the block of current by NA in a background of baclofen. Baclofen completely occluded response to subsequent application of NA. Statistical significance, ***p<0.001. Dashed line indicates initial current level. See **Figure 7—source data 1**.

The online version of this article includes the following source data for figure 7:

**Source data 1.** Source data for **Figure 7B, C**.

Since both NA and baclofen inhibit NALCN in CWC, we asked whether these two receptors act on independent or overlapping populations of NALCN channels. In these experiments, one set of control recordings assessed the magnitude of block by NA of low-$Ca^{2+}$-evoked inward current. Interleaved with these recordings were experiments in which baclofen was applied to reduce the NALCN current, and then NA subsequently applied in the continued presence of baclofen (**Figure 7A–C**). If the two agonists act on independent populations of NALCN channels, NA should suppress inward current similar to that seen in control experiments. However, as shown in **Figure 7**, NA had virtually no blocking action of inward current following application of baclofen (0.1 mM $Ca^{2+}$ = −164.28 ± 20.50 pA, baclofen = −53.33 ± 14.73 pA. NA = −46.94 ± 16.50 pA, N = 9, F = 41.28, p<0.0001, repeated-measures one-way ANOVA; post hoc Tukey's test, 2 mM $Ca^{2+}$ vs 0.1 mM $Ca^{2+}$ p=0.002, 0.1 mM $Ca^{2+}$ vs baclofen p=0.005, 0.1 mM $Ca^{2+}$ vs NA p=0.002, baclofen vs NA p=0.84). We also examined the percentage block of 0.1 mM $Ca^{2+}$ current in baclofen, or NA alone, compared to the block of current by NA in a background of baclofen. Baclofen completely occluded the response to subsequent application of NA (percent block: baclofen 70.61 ± 7.18, NA 56.18 ± 9.5, additional block 4.35 ± 3.25, n = 9, $F_{(2,24)}$ = 23.83, p<0.0001, one-way ANOVA; post hoc Tukey's test, baclofen vs NA p=0.34, baclofen vs addition p<0.0001, NA vs addition p<0.0001). These data suggest that α2 receptors target the same set of NALCN channels as $GABA_B$ receptors, implying close apposition of the two receptors and the NALCN channel complex.

## Noradrenergic effect not mediated by control of cAMP levels

The α2 receptor is a Gi/o GPCR, and thus, it is possible that inhibition of adenylyl cyclase and thus reduction of cAMP levels might mediate the action of NA on NALCN. To test this idea, CWCs were recorded in the same solutions used in the previous section in order to block $K^+$ channels and other non-NALCN contributions to current. Then NA (100 µM) was puffed onto the recorded neuron, which suppresses NALCN and generates a net outward current. After recording control responses, 1 mM 8-Br-cAMP was washed into the bath, and NA responses again recorded. The average response amplitudes were 23.9 ± 3.5 pA in control and 27.8 ± 5.7 pA in 8-Br-cAMP (n = 7 cells, p=0.1147, paired t-test). On the assumption that 8-Br-cAMP would effectively saturate cAMP actions in the neurons, the absence of an effect on the NA response suggests that inhibition cAMP production does not mediate NA action on NALCN, pointing to the possibility of direct effect of G-proteins on the channel.

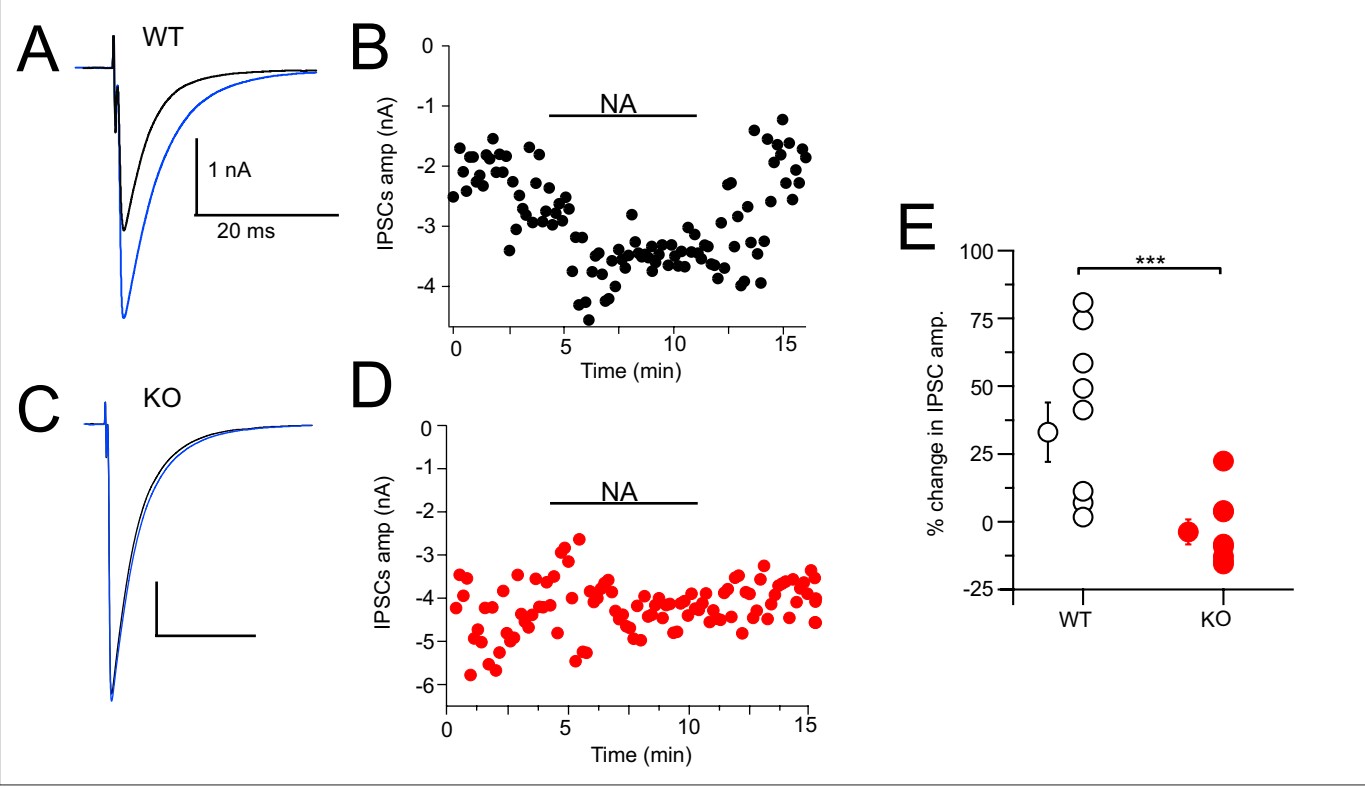

**Figure 8.** Noradrenaline (NA) enhances cartwheel cell (CWC)-mediated IPSCs in wildtype (WT) but not in NALCN cKO cells. (**A**) Example average trace of evoked IPSCs recorded from a CWC with a CsCl pipette fill, before (black) and after (blue) bath application of 10 μM NA. (**B**) Diary plot of effect of NA application (applied during black bar) of the IPSCs amplitude in cell (**A**). (**C**) Example average trace of evoked IPSCs recorded from a CWC as in (**A**), but from a NALCN cKO mouse. Data shown before (black) and after (blue) bath application of 10 μM NA. (**D**) Diary plot of effect of NA application in (**C**). (**E**) Summary plot of percent change of IPSCs amplitude after NA application in WT and knockout (KO) mice (WT, N = 8; KO, N = 10). See *Figure 8—source data 1*.

The online version of this article includes the following source data for figure 8:

**Source data 1.** Source data for *Figure 8E*.

## Noradrenergic enhancement of glycine release

CWCs mediate synaptic inhibition through the release of glycine onto fusiform principal cells as well as onto other CWCs in the DCN (*Mancilla and Manis, 2009*; *Roberts and Trussell, 2010*; *Kuo and Trussell, 2011*; *Apostolides and Trussell, 2014*). NA has been shown to enhance feedforward inhibition by suppressing background spiking activity (*Kuo and Trussell, 2011*; *Lu and Trussell, 2016*). We tested more directly the effect of NA on synaptic inhibition and the role of NALCN by electrically evoking inhibitory postsynaptic currents (IPSCs) while recording from CWC. Bipolar or glass stimulation electrodes were positioned in the DCN molecular layer in order to stimulate CWC axons. The resulting glycinergic IPSCs were significantly enhanced upon bath application of 10 μM NA (Ctrl = –1.37 ± 0.31 nA, NA = –1.82 ± 0.43 nA, % change = 32.99% [increase], N = 8, *Figure 8A and B*). This average increase ranged widely from 2 to 81%, suggesting some cells received glycinergic input from CWC that were not spontaneously active and therefore not affected by NA. The experiment was repeated on NALCN cKO mice, and there was no significant increase in IPSC amplitude upon addition of NA (KO = –2.17 ± 0.54 nA, NA = –2.08 ± 0.55 nA, % change = 3.77% [decrease], N = 10, p=0.0011; *Figure 8C and D*). Thus, NA-mediated increase in the strength of inhibition (*Kuo and Trussell, 2011*) is likely mediated by NALCN.

## Discussion

### Mechanism of adrenergic modulation

The primary effect of noradrenergic modulation of the CWC is to enhance the strength of inhibitory synaptic transmission (*Kuo and Trussell, 2011*). Modulation by NA acts simultaneously on all the CWC neurons synapses by impacting spike generation and thereby the degree of frequency-dependent synaptic depression. We show here that NALCN is the ion channel that mediates this effect on firing. $\alpha 2$ receptors are known to mediate inhibition of firing through enhancement of $K^+$ channel activity, and GIRK channels play a prominent role in these effects (*Williams et al., 1985*; *Aghajanian and Wang, 1986*; *Arima et al., 1998*; *Li and van den Pol, 2005*). However, in CWC, almost none of the effect of NA was blocked by the $K^+$ channel blocker $Ba^{2+}$. Indeed, since in NALCN cKOs the NA-induced effects on spiking, outward currents, or block of inward currents were all eliminated, it is almost certain that NALCN and not $K^+$ channels play the key role in noradrenergic modulation in CWC. A caveat is that we only examined ionic currents and voltage changes measurable with a somatic electrode. NA effects in distal dendrites may have been undetectable at the soma.

Several intriguing aspects of NA action are revealed in this work when considered together with our previous study (*Kuo and Trussell, 2011*). First, the action of NA on feedforward inhibition is apparently quite specific to NALCN. No effect of NA was observed on parallel fiber EPSCs in CWC or CWC IPSCs in fusiform cells recorded during paired voltage-clamp recordings (*Kuo and Trussell, 2011*). This suggests that the mechanism for control of inhibitory strength is optimized to act globally on the CWC's synapses by controlling spontaneous firing. Presumably, $\alpha 2$ receptors are localized in such a way to affect membrane potential and rheobase at the soma and/or at the axon initial segment, although localization studies will be required to address this uncertainty. Despite the DCN's dense innervation from noradrenergic fibers (*Kromer and Moore, 1980*; *Klepper and Herbert, 1991*), we do not know the morphology of noradrenergic synapses in the DCN. However, if NA release is through 'volume transmission,' it may be that noradrenergic modulation acts on groups of CWCs and all their synapses. Second, although CWCs have GIRK channels, and $\alpha 2$ activates GIRK channels in some other neuronal cell types (*Arima et al., 1998*; *Paladini and Williams, 2004*; *Li and van den Pol, 2005*), the CWC has selected for a mechanism in which an inhibitory action is associated with a decrease in a tonic membrane conductance. The effect on membrane potential of this decrease in conductance is small, ~2 mV. Yet, we show that that shift alone is sufficient to profoundly inhibit spontaneous firing and thereby increase the strength of inhibitory synapses. Interestingly, an inverse mechanism of modulation was described in hippocampus, in which oxytocin increases spontaneous firing of interneurons and thus decreases inhibitory synaptic strength (*Owen et al., 2013*). In this case, oxytocin acts by decrease of a resting inhibitory conductance mediated by Kv7 channels (*Tirko et al., 2018*). Thus, modulation by inhibitory or excitatory conductance decrease provides a means for global control of inhibitory synaptic strength in the CNS.

### Convergence of GABAergic and adrenergic systems

Unlike $\alpha 2$ receptors, the actions of GABA$_B$ receptors were more complex as GABA$_B$ receptors both inhibited NALCN and activated GIRK channels, as previously observed in substantia nigra neurons (*Philippart and Khaliq, 2018*). Comparisons of the magnitude of NA action in control recordings and in a background of baclofen indicated that the two modulators acted on a common population of NALCN channels. While activation of $\alpha 2$ receptors may lead to inhibition of adenylyl cyclase (*Dohlman et al., 1987*), we did not observe an effect of 8-Br-cAMP on NA responses, arguing against a role for cAMP. If gating of the channels proceeds directly by membrane-delimited actions of G-proteins, it is likely that a pool of GABA$_B$ and $\alpha 2$ receptors are in relatively close apposition to one another and to NALCN. However, unlike GABA$_B$ receptors, $\alpha 2$ receptors did not appreciably activate GIRK channels. Altogether, these results suggest two distinct pools of GABA$_B$ receptors in the somatodendritic membrane: one associated with GIRK and one with NALCN. In addition, a third pool is localized in CWC nerve terminals. There, baclofen suppresses release of action potential-independent exocytosis (*Apostolides and Trussell, 2013*), suggesting modulation by GABA$_B$ receptors of $Ca^{2+}$ channels or exocytic proteins. Regarding the somatodendritic populations, immunolocalization with electron microscopy indicated that GABAB1 subunits are concentrated at the base of dendritic spines of CWC, rather than associated with GABAergic synapses (*Luján et al., 2004*), with the suggestion that they

respond to 'spillover' of GABA. It remains unclear which ion channel this particular dendritic GABA$_B$ population modulates.

## Auditory processing

The DCN is the site of diverse forms of neuromodulation mediated by G-protein-coupled receptors (*Trussell and Oertel, 2018*; *Trussell, 2019*), in which transmitters act at presynaptic membrane to regulate exocytosis (*Tang and Trussell, 2015*; *Tang and Trussell, 2017*), at the axon initial segment to control spike pattern (*Bender et al., 2010*), at somatodendritic membrane to boost firing (*Tang and Trussell, 2017*), or at dendritic spines to control postsynaptic efficacy (*He et al., 2014*). By restricting α2 modulation to the spike-generation mechanism and not presynaptic release zones, the transmitter is able to enhance IPSC amplitude while simultaneously reducing background inhibitory 'noise,' thus enhancing the salience of a given interneuron's impact. CWCs are activated by parallel fibers conveying multisensory input to DCN fusiform cells. If, consistent with other studies (*Berridge and Waterhouse, 2003*), NA is released during brain states associated with wakefulness, this enhanced signal-to-noise mediated by NA would serve to more precisely sculpt the output of DCN based on such multisensory signals. Moreover, since individual CWC-fusiform synaptic pairs apparently receive excitatory input from different populations of parallel fibers (*Roberts and Trussell, 2010*), such enhanced inhibition might serve a more precise computational role than a standard feedforward inhibitory mechanism. NALCN therefore appears to play a potent role in selective sensory filtering during heightened states of vigilance.

# Materials and methods

**Key resources table**

| Reagent type (species) or resource | Designation | Source or reference | Identifiers | Additional information |
|---|---|---|---|---|
| Strain, strain background (*Mus musculus*) | C57BL/6J | Jackson Laboratory | RRID:JAX:000664 | |
| Strain, strain background (*M. musculus*) | GlyT2-EGFP, Tg(Slc6a5-EGFP)1Uze | MGI, *Zeilhofer et al., 2005* | RRID:MGI:J:145521 | |
| Strain, strain background (*M. musculus*) | B6(Cg)-*Nalcn*$^{tm1c(KOMP)Wtsi}$/DrenJ | MGI and Jackson Laboratory | RRID:JAX_030718 | |
| Strain, strain background (*M. musculus*) | Tg(Slc6a5-cre)KF109Gsat | MGI | RRID:MGI:4367229 | |
| Chemical compound, drug | Strychnine hydrochloride | Sigma | Cat# S8753 | |
| Chemical compound, drug | SR-95531 hydrobromide | Tocris Bioscience | Cat# 1262 | |
| Chemical compound, drug | NBQX disodium salt | Tocris Bioscience | Cat# 1044 | |
| Chemical compound, drug | (+)-MK-801 hydrogen maleate | Sigma | Cat# M107 | |
| Chemical compound, drug | L-(-)-Norepinephrine (+)-bitartrate salt monohydrate | Sigma | Cat# A9512 | |
| Chemical compound, drug | (±)-Norepinephrine (+)-bitartrate salt | Sigma | Cat# A0937 | |
| Chemical compound, drug | Idazoxan hydrochloride | Sigma | Cat# I6138-100MG | |
| Chemical compound, drug | Apamin | Alomone Labs | Cat# STA-200 | |

*Continued on next page*

*Continued*

| Reagent type (species) or resource | Designation | Source or reference | Identifiers | Additional information |
|---|---|---|---|---|
| Chemical compound, drug | D-AP5 | Tocris Bioscience | Cat# 0106 | |
| Chemical compound, drug | Barium chloride | Sigma | Cat# 202738 | |
| Software, algorithm | pClamp 10 | Molecular Devices | RRID:SCR_011323 | |
| Software, algorithm | Igor Pro 8 | WaveMetrics | RRID:SCR_000325 | |
| Software, algorithm | NeuroMatic | *Rothman and Silver, 2018*; DOI:10.3389/fninf.2018.00014 | RRID:SCR_004186 | |
| Software, algorithm | Axograph | Axograph | RRID:SCR_014284 | |
| Software, algorithm | Prism 9 | GraphPad | RRID:SCR_002798 | |
| Software, algorithm | Excel | Microsoft | RRID:SCR_016137 | |
| Software, algorithm | Affinity Designer | Serif | RRID:SCR_016952 | |

## Animals

All procedures were approved by the Oregon Health and Science University's Institutional Animal Care and Use Committee. C57BL/6J, GlyT2-EGFP mice (*Zeilhofer et al., 2005*; *Ngodup et al., 2020*), and NALCN cKO mice of either sex, postnatal days (P) 17–40 were used in this study. GlyT2-GFP mice were backcrossed into the C57BL/6J and maintained as heterozygous. As global knockout of NALCN is lethal, we generated a glycinergic neuron-specific knockout by crossing NALCN$^{flox/flox}$ mice (B6(Cg)-*Nalcn$^{tm1c(KOMP)Wtsi}$*/DrenJ) with GlyT2-Cre mice (Tg(Slc6a5-cre)KF109Gsat), resulting in NALCN-$^{flox}$;GlyT2-Cre offspring. The F1 litters were back-crossed with NALCN$^{flox/flox}$ mice to generate NALCN-$^{flox/flox}$;GlyT2-Cre that lack NALCN expression in glycinergic cells. NALCN cKO mice were smaller than age-matched WT litters but had normal hearing as established by ABR.

## Brain-slice preparation

Animals were anesthetized with isoflurane and decapitated. The brain was quickly removed and placed into ice-cold sucrose cutting solution. Sucrose solution contained (in mM) 76 NaCl, 26 NaHCO$_3$, 75 sucrose, 1.25 NaH$_2$PO$_4$, 2.5 KCl, 25 glucose, 7 MgCl$_2$, and 0.5 CaCl$_2$, bubbled with 95% O$_2$:5% CO$_2$ (pH 7.8, 305 mOsm). Coronal slices containing DCN were cut at 210 µm in ice-cold sucrose solution on a vibratome (VT1200S; Leica Microsystems, Wetzlar, Germany or 7000smz-2; Campden Instruments, Loughborough, UK). Slices were transferred into standard artificial cerebrospinal fluid (ACSF) containing (in mM) 125 NaCl, 20 NaHCO$_3$, 1.2 KH$_2$PO$_4$, 3 HEPES, 2.1 KCl, 20 glucose, 1 MgCl$_2$, 1.2 CaCl$_2$, 2 Na-pyruvate, and 0.4 Na L-ascorbate, bubbled with 95% O$_2$:5% CO$_2$ (pH 7.4, 300–310 mOsm). Slices were recovered at 34°C for 40 min and were maintained at room temperature until recording.

## Electrophysiology

Slices were transferred to a recording chamber and perfused with standard ACSF at 3 ml/min and maintained at 31–34°C with an in-line heater (TC-324B; Warner Instruments, Hamden, CT). Cells were viewed using an upright microscope (BX51WI; Olympus, Tokyo, Japan) with a ×60 objective, equipped with custom-made infrared Dodt contrast optics, CCD camera (Retiga 2000R; QImaging, Surrey, Canada), and fluorescence optics. All recordings were collected from CWCs of the DCN. CWCs were targeted by their location in the molecular and fusiform cell layers of DCN, and by their round soma (*Kim and Trussell, 2007*). Identification was then confirmed by their distinctive firing pattern (simple or complex spikes) (*Golding and Oertel, 1997*). In slices from GlyT2-EGFP, glycinergic cells in the DCN were identified by their GFP expression. In some experiments, 0.1% biocytin (B1592; Thermo Fisher Scientific, Waltham, MA) was added to the pipette solution for post hoc identification of CWCs. Recording pipettes were pulled from 1.5 mm OD, 0.84 mm ID borosilicate glass (1B150-F; World Precision Instruments, Sarasota, FL) to a resistance of 2–4 MΩ using a horizontal puller (P-97 or P-1000; Sutter Instruments, Novato, CA). In most experiments, internal recording solution contained (in mM) 113 K gluconate, 2.75 MgCl$_2$, 1.75 MgSO$_4$, 0.1 EGTA, 14 Tris$_2$-phosphocreatine, 4 Na$_2$-ATP,

0.3 Tris-GTP, 9 HEPES with pH adjusted to 7.25 with KOH, mOsm adjusted to 290 with sucrose ($E_{Cl}$, –84 mV). For voltage clamp to isolate NALCN current, internal solution contained (in mM) 87 $CsMeSO_3$, 18 CsCl, 5 CsF, 10 TEA-Cl, 10 HEPES, 5 EGTA, 5 Mg-ATP, 0.3 $Na_2$-GTP, 13 di-Na phosphocreatine, 2 QX-314 (pH 7.25, 295 mOsm). For a few voltage-clamp experiments, we used an internal solution containing (in mM) 103 CsCl, 10 TEA-Cl, 2.75 $MgCl_2$, 9 HEPES, 0.1 EGTA, 0.3 Tris-GTP, 14 $Tris_2$-phosphocreatine, 4 $Na_2$-ATP, 3.5 QX-314 (pH adjusted to 7.2 with CsOH). Puff application of agonists and antagonists was delivered through a picospritzer (Picospritzer III; Toohey Company, Fairfield, NJ), at 7–10 psi, with borosilicate glass capillaries. NA or baclofen applications were at 100 µM and 50–100 ms in duration. The puff pipette was placed around 100 µm from the soma of the recorded cell to avoid mechanical disturbance.

Cell-attached (voltage-clamp) recordings were made using normal extracellular solution. Whole-cell patch-clamp recordings were made using a Multiclamp 700B amplifier and pCLAMP 10 software (Molecular Devices, Sunnyvale, CA). Signals were digitized at 20–40 kHz and filtered at 10 kHz by Digidata 1440A (Molecular Devices). In voltage clamp, cells were held at –65 mV, with access resistance 5–30 MΩ compensated to 40–60%. In current clamp with control solutions, the resting membrane potential was maintained at –60 to –70 mV with bias current. To isolate NALCN currents, synaptic blockers, NBQX (10 µM), MK-801 (10 µM), SR-95531 (10 µM) or picrotoxin (100 µM), strychnine (0.5 µM), apamin (100 nM), and $BaCl_2$ (200 µM) were added to the bath solution. In those experiments, shifts in $Ca^{2+}$ from 2 mM to 0.1 mM were accompanied by a shift in $Mg^{2+}$ from 1 mM to 3 mM. To record evoked IPSCs, CWCs were stimulated with brief voltage pulses (100 µs) using a stimulus isolation unit (Iso-Flex; A.M.P.I, Jerusalem, Israel) via a bipolar or glass microelectrode placed in the molecular layer of the DCN.

## Pharmacology

All drugs in the slice experiments were bath applied. Receptor antagonists used in this study included NBQX (AMPA receptors; Sigma-Aldrich, St. Louis, MO), MK-801 (NMDA receptors; Sigma-Aldrich), SR-95531 ($GABA_A$R; Tocris Bioscience, Bristol, UK), and strychnine (glycine receptors; Sigma-Aldrich).

## Auditory brainstem responses

ABRs were acquired from C57B6/J and NALCN cKO mice between P40-50. Mice were anesthetized with a dose of 80 mg/kg ketamine:16 mg/kg xylazine. ABRs were recorded differentially with electrodes at the vertex and pinna with ground at the base of the tail. Tone pips (5 ms, 0.5 ms rise/fall with 4 ms steady-state plateau) were generated digitally with a PXI-4461 card (National Instruments, Austin, TX), amplified (SA1; Tucker-Davis Technologies, Alachua, FL) and delivered using a compact, close-field sound system consisting of two CUI earphones and an electric microphone (FG-23329-P07; Knowles, Itasca, IL) coupled to a probe tube (*Buran, 2015*; *Hancock et al., 2015*; *Buran et al., 2020*). Tone levels were incremented in 5 dB steps from 0 to 90 dB SPL. ABRs were recorded in a sound-proof booth using a Grass P511 amplifier and digitized using a National Instruments PXI 4661 card. ABR threshold was obtained for each animal. ABR peaks and thresholds were identified by visual inspection of the waveforms in a custom-written analysis program (*Buran, 2015*). Body temperature was maintained between 36 and 37°C using a homeothermic blanket (Harvard Apparatus, Cambridge, MA).

## Experimental design and statistical analyses

Electrophysiological data were analyzed using pClamp 10.4 software (Molecular Devices), Axograph, or IGOR Pro v6.3 or v8 (WaveMetrics, Lake Oswego, OR) and NeuroMatic (*Rothman and Silver, 2018*). Figures were made using IGOR Pro, Affinity Designer, and Adobe Illustrator. Statistics were performed in IGOR Pro, Axograph, Python, Microsoft Excel, or Prism 9 (GraphPad, San Diego, CA). Student's *t*-test and ANOVA were used to compare the means when datasets were normally distributed. Otherwise, non-parametric tests were employed. Error bars are represented as mean ± SEM unless otherwise stated.

## Acknowledgements

This work was supported by NIH grants R35NS116798 and DC004450 to LOT and JSPS KAKENHI Grant Number 22K06838 to TI. We thank Dr. Brad Buran for help with ABRs, and Dr. John Williams and the members of the Trussell Lab for comments.

# Additional information

## Funding

| Funder | Grant reference number | Author |
|---|---|---|
| National Institute of Neurological Disorders and Stroke | R35NS116798 | Laurence O Trussell |
| National Institute on Deafness and Other Communication Disorders | DC004450 | Laurence O Trussell |
| Japan Society for the Promotion of Science | KAKENHI 22K06838 | Tomohiko Irie |

The funders had no role in study design, data collection and interpretation, or the decision to submit the work for publication.

## Author contributions

Tenzin Ngodup, Tomohiko Irie, Conceptualization, Data curation, Formal analysis, Validation, Investigation, Methodology, Writing – original draft, Writing – review and editing; Seán P Elkins, Investigation; Laurence O Trussell, Conceptualization, Resources, Data curation, Formal analysis, Supervision, Funding acquisition, Validation, Investigation, Visualization, Writing – original draft, Project administration, Writing – review and editing

## Author ORCIDs

Laurence O Trussell (iD) https://orcid.org/0000-0003-1171-2356

## Ethics

All procedures were approved by the Oregon Health and Science University's Institutional Animal Care and Use Committee, protocol #IP00000952, and were conducted in strict accordance with those guidelines.

Reviewer #1 (Public Review): https://doi.org/10.7554/eLife.89520.3.sa1
Reviewer #2 (Public Review): https://doi.org/10.7554/eLife.89520.3.sa2
Reviewer #3 (Public Review): https://doi.org/10.7554/eLife.89520.3.sa3
Author Response https://doi.org/10.7554/eLife.89520.3.sa4

# Additional files

## Supplementary files

• MDAR checklist

## Data availability

All data generated or analyzed during this study are included in the manuscript and supporting file.

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
