## [Editor Report · eLife assessment]

This paper reports the **fundamental** discovery of adrenergic modulation of spontaneous firing through the inhibition of the Na^+^ leak channel NALCN in cartwheel cells in the dorsal cochlear nucleus. This study provides unequivocal evidence that the activation of α-2 adrenergic or GABA-B receptors inhibit NALCN currents to reduce neuronal excitability. The evidence supporting the conclusions is **exceptional**, the electrophysiological data is high quality, and the experimental design is rigorous.

---

## [Referee Report · Reviewer #1 (Public Review)]

This study uses electrophysiological techniques in vitro to address the role of the Na+ leak channel NALCN in various physiological functions in cartwheel interneurons of the dorsal cochlear nucleus. Comparing wild type and glycinergic neuron-specific knockout mice for NALCN, the authors show that these channels (1) are required for spontaneous firing, (2) are modulated by noradrenaline (NA, via alpha2 receptors) and GABA (through GABAB receptors), (3) how the modulation by NA enhances IPSCs in these neurons.

This work builds on previous results from the Trussell's lab in terms of the physiology of cartwheel cells, and from other labs in terms of the role of NALCN channels, that have been characterized in more and more brain areas somewhat recently; for this reason, this study could be of interest for researchers that work in other preparations as well. The general conclusions are strongly supported by results that are clearly and elegantly presented.

In this revised submission, the authors addressed all my questions. This is very interesting work that could be of interest for researchers working in other brain areas as well.

---

## [Referee Report · Reviewer #2 (Public Review)]

This is a very interesting paper with several important findings related to the working mechanism of the cartwheel cells (CWC) in the dorsal cochlear nucleus (DCN). These cells generate spontaneous firing that is inhibited by the activation of α2-adrenergic receptors, which also enhances the synaptic strength in the cells, but the mechanisms underlying the spontaneous firing and the dual regulation by α2-adrenergic receptor activation have remained elusive. By recording these cells with the NALCN sodium-leak channel conditionally knocked, the authors discovered that both the spontaneous firing and the regulation by noradrenaline (NA) require NALCN. Mechanistically, the authors found that activation of the adrenergic receptor or GABAB receptor inhibits NALCN. Interestingly, these receptor activations also suppress the low [Ca2+] "activation" of NALCN currents, suggesting crosstalk between the pathways. The finding of such dominant contribution of the NALCN conductance to the regulation of firing by NA is somewhat surprising considering that NA is known to regulate K+ conductances in many other neurons.

The studies reveal the molecular mechanisms underlying well known regulations of the neuronal processes in the auditory pathway. The results will be important to the understanding of auditory information processing in particular, and, more generally, to the understanding of the regulation of inhibitory neurons and ion channels. The results are convincing and are clearly presented.

In this revision, the authors have satisfactorily addressed all my previous comments.

---

## [Referee Report · Reviewer #3 (Public Review)]

The study by Ngodup and colleagues describes the contribution of sodium leak NALCN conductance on the effects of noradrenaline on cartwheel interneurons of the DCN. The manuscript is very well-written and the experiments are well-controlled. The scope of the study is of high biological relevance and recapitulates a primary finding of the Khaliq lab (Philippart et al., eLife, 2018) in ventral midbrain dopamine neurons, that Gi/o-coupled receptors inhibit NALCN current to reduce neuronal excitability. Together these studies provide unequivocable evidence for NALCN as a downstream target of these receptors.

In re-review of this study, the authors have addressed the concerns sufficiently. This is a very nice study.

---

## [Author Response]

The following is the authors’ response to the original reviews.

**eLife assessment**
This paper reports the fundamental discovery of adrenergic modulation of spontaneous firing through the inhibition of the Na+ leak channel NALCN in cartwheel cells in the dorsal cochlear nucleus. This study provides unequivocal evidence that the activation of alpha-2 adrenergic or GABA-B receptors inhibit NALCN currents to reduce neuronal excitability. The evidence supporting the conclusions is compelling, the electrophysiological data is high quality and the experimental design is rigorous.
**Public Reviews:**

**Reviewer #1 (Public Review):**
This study uses electrophysiological techniques in vitro to address the role of the Na+ leak channel NALCN in various physiological functions in cartwheel interneurons of the dorsal cochlear nucleus. Comparing wild type and glycinergic neuron-specific knockout mice for NALCN, the authors show that these channels (1) are required for spontaneous firing, (2) are modulated by noradrenaline (NA, via alpha2 receptors) and GABA (through GABAB receptors), (3) how the modulation by NA enhances IPSCs in these neurons.This work builds on previous results from the Trussell's lab in terms of the physiology of cartwheel cells, and from other labs in terms of the role of NALCN channels, that have been characterized in more and more brain areas somewhat recently; for this reason, this study could be of interest for researchers that work in other preparations as well. The general conclusions are strongly supported by results that are clearly and elegantly presented.I have a few comments that, in my opinion, might help clarify some aspects of the manuscript.1. It is mentioned throughout the manuscript, including the abstract, that the results suggest a closed apposition of NALCN channels and alpha2 and GABAB receptors. From what I understand, this conclusion comes from the fact that GABAB receptors activate GIRK channels through a membrane-delimited mechanism. Is it possible that these receptors converge on other effectors, for example adenylate cyclase (see https://www.ncbi.nlm.nih.gov/pmc/articles/PMC6374141/).

We have now tested the role of adenylyl cyclase modulation in the control of NALCN, by saturating the cells with a cAMP analogue 8-Br-cAMP and found no effect on the NA response. These data are included in the paper. While further experiments are necessary, these results argue in favor of a direct gating by G-proteins.

1. In Figure 2G, the neurons from NALCN KO mice appear to reach a significantly higher frequency than those from WT (figure 2E, 110 vs. 70 spikes/s). Was this higher frequency a feature of all experiments? The results mention a rundown of peak firing rate due to whole-cell dialysis, but, from what I understand, the control conditions should be similar for all experiments.

The peak firing rates in control solutions for WT and KO CWC are not statistically different.

1. Also in Figure 2, the firing patterns for neurons from WT and NALCN KO mice appear to be quite different, with spikes appearing to be generated during the hyperpolarization of the bursts in the second half of the current step for WT neurons but always during the depolarization in KO neurons. Was this always the case? If so, could NALCN channels be involved in this type of firing? Along these lines, it would be interesting to show an example of a firing pattern of neurons from WT mice in the presence of NA, which inhibits NALCN channels.

The specific pattern of spikes in CWC is quite variable from trial-to-trial or cell-to-cell, as it is dependent on multiple CaV and calcium dependent K channels subtypes, and is not dependent on the genotypes used here. The primary effects observed in the KO are in background firing and sensitivity to NA, both reflected alterations in rheobase. The firing pattern example requested was shown in the raster plot of fig 2B2.

1. It might be interesting to discuss how the hyperpolarization induced by the activation of GIRK channels and inhibition of NALCN channels could have different consequences due to their opposite effect on the input resistance.

We considered this as a point of discussion, but decided that making sense of it would depend on assumptions about the location of the channels (dendritic vs somatic, distance to AIS) that we do not have data for. For example, a dendritic increase in resistance through NALCN block, leading to a hyperpolarization of the soma, might have actions similar to a somatic hyperpolarizing conductance increase by GIRK, as far as the voltage at the AIS is concerned.

**Reviewer #2 (Public Review):**
This is a very interesting paper with several important findings related to the working mechanism of the cartwheel cells (CWC) in the dorsal cochlear nucleus (DCN). These cells generate spontaneous firing that is inhibited by the activation of α2-adrenergic receptors, which also enhances the synaptic strength in the cells, but the mechanisms underlying the spontaneous firing and the dual regulation by α2-adrenergic receptor activation have remained elusive. By recording these cells with the NALCN sodium-leak channel conditionally knocked, the authors discovered that both the spontaneous firing and the regulation by noradrenaline (NA) require NALCN. Mechanistically, the authors found that activation of the adrenergic receptor or GABAB receptor inhibits NALCN. Interestingly, these receptor activations also suppress the low [Ca2+] "activation" of NALCN currents, suggesting crosstalk between the pathways. The finding of such dominant contribution of the NALCN conductance to the regulation of firing by NA is somewhat surprising considering that NA is known to regulate K+ conductances in many other neurons.The studies reveal the molecular mechanisms underlying well known regulations of the neuronal processes in the auditory pathway. The results will be important to the understanding of auditory information processing in particular, and, more generally, to the understanding of the regulation of inhibitory neurons and ion channels. The results are convincing and are clearly presented.
**Reviewer #3 (Public Review):**
The study by Ngodup and colleagues describes the contribution of sodium leak NALCN conductance on the effects of noradrenaline on cartwheel interneurons of the DCN. The manuscript is very well-written and the experiments are well-controlled. The scope of the study is of high biological relevance and recapitulates a primary finding of the Khaliq lab (Philippart et al., eLife, 2018) in ventral midbrain dopamine neurons, that Gi/o-coupled receptors inhibit NALCN current to reduce neuronal excitability. Together these studies provide unequivocable evidence for NALCN as a downstream target of these receptors. There are no major concerns. I have only minor suggestions:Minor1. As introduced in the introduction, NALCN is inhibited by extracellular calcium which has led to some discourse of the relevance of NALCN when recorded in 0.1 mM calcium. A strength of this study is the effect of NA on NALCN is recorded in physiological levels of calcium (1.2 mM). I suggest including the concentration of extracellular calcium in the aCSF in the Results section instead of relying on the reader to look to the Methods.

Done.

1. It would be interesting to include the basal membrane properties of the KO compared to wildtype, including membrane resistance and resting membrane potential. From the example recording in Figure 2, one might think that the KOs have lower membrane resistance, so it is interesting that the 2 mV hyperpolarization produced similar effects on rheobase. In addition, from the example in Figure 2G, it appears that NA has an effect on firing frequency with large current injection in the KO. Is this true in grouped data and if so, is there any speculation into how this occurs?

We have included in the text a comparison of the input resistance in WT and KO. These were not different. This should not be too surprising given the wide range of values between animals, and the necessity to compare populations. Measurements of resting potential are complicated by the fact that CWC are normally spontaneously active.As was discussed in the text, peak firing frequency declined with time during recording in both control and KO, necessitating normalization as shown in Fig 2E-H.

1. Please expand on the rationale for why GABAB and alpha2 must be physically close to NALCN. To my knowledge, the mechanism by which these receptors inhibit NALCN is not known. Must it be membrane-delimited?

Given the known membrane delimited modulation of GIRK by GABAB, and that alpha2 and GABAB receptors appear to share the same population of NALCN channels, and that alpha2 receptors do not appear to target GIRK channels, we felt the simplest explanation would be coupling through G-proteins, with spatial segregation of different receptor/channel pools providing the means for separating GIRK and NALCN effects. Given that the alpha2 receptor is a Gi/o GPCR, we have now included in the revision new experiments using 8-Br-cAMP, as discussed above. These showed no effect on the NA response, consistent with a direct effect membrane delimited of G-proteins. We acknowledge however that further experiments are warranted.

**Reviewer #1 (Recommendations For The Authors):**
1. I suggest labeling the voltage traces in Figure 2 with WT and KO for easier comprehension; in addition, I suggest adding the average data to the plots in Figure 2, as in Figure 2-supplementary Figure 1 panel F.

We have added the figure labels as requested. We chose not to add the average data as we noticed that averaging the full FI plots led to a smearing of the curves and a distortion in the apparent rheobase. Thus, we instead measured the rheobase for individual cells and report their average.

1. For readers that are not familiar with the field, more details should be given about the electrical stimulation to evoke IPSCs in cartwheel cells, and what they represent.

Done.

1. The methods should mention if and how the concentrations of divalents were adjusted in the experiments with 0.1 extracellular Ca2+

Done.

**Reviewer #2 (Recommendations For The Authors):**
I only have several minor comments.1. The total lack of spontaneous firing in CWCs in the NALCN KO (Fig. 1) is interesting and provides an opportunity to probe the in vivo function of such spontaneous firing. Besides being a little smaller, do the mutant mice have any sign of abnormality in sound signal processing?

Figure 1 – Figure supplement 1 showed that there are no effects on auditory brainstem responses in the KO.

1. Figs. 3&4 (and several other figures with voltage-clamp recordings), a line indicating zero current level would be useful.

Done

1. page 7, "Outward current generated by suppression of NALCN": it might be better to state as "Outward response generated by suppression of NALCN", as the authors correctly pointed out that the NA-induced apparently outward current response is largely a result of an inhibition of NALCN-mediated inward Na+ current. One way to clarify this might be to record at the Nernst potential of K+ to isolate the contribution of Na+ currents (unclear if K+- or Cs+-based pipette was used in the experiment in Fig 3).

Text has been modified.

1. Figs. 5,6&7: do the dashed lines indicate initial current level or zero current level?

Initial current. See legends.

1. The labeling of some of the bar graphs can be made more clear. For example, in Fig. 2K, the right two columns should be labeled as WT as well. Fig. 3C & Fig. 4C, the left two columns should be labeled as WT and the right two as KO.

Added labels to Fig 2 as requested.

1. Figs. 5-7: The suppression of low extracellular [Ca2+]-induced NALCN-dependent current by NA and baclofen is very interesting. As the tonic inhibition of NALCN by extracellular Ca2+ is likely through a Ca2+-sensing GPCR (CaSR) and G-proteins (lowering [Ca2+] releases the inhibition and generates inward current) (Lu et al. 2010), the action of NA and baclofen may all converge onto the same G-protein dependent pathway of the Ca2+-sensing receptor. I'd include this in the discussion to provide a potential mechanistic explanation of the interesting observation.

This is indeed an interesting idea. We prefer not to discuss here, as (1) the source of Ca2+ sensitivity of the channel seems to be controversial (Chua et al 2020), and (2) the effect of Ca2+ reduction is enormously slower than the effect of the modulators (Fig 5-7), implying distinct mechanisms.

**Reviewer #3 (Recommendations For The Authors):**
Typos/general comments1. Figure 2 would be easier to comprehend with WT and KO labels as in the other figures. Done1. Page 11, size of the IPSCs in NA is missing the minus sign.

Corrected.

1. Is the y-axis correct on Figure 8B? This looks like it is doubling the size of the IPSC.

Thank you for catching this mistake. The formula used to calculate % change was in error. We have corrected all the data analysis in the figure, which fortunately did not change the conclusion. Regarding the axis, note that the measurement was % change, not ratio of drug vs control.